# Application of Pomace Powder of Black Carrot as a Natural Food Ingredient in Yoghurt

**DOI:** 10.3390/foods13071130

**Published:** 2024-04-08

**Authors:** Florina Stoica, Roxana Nicoleta Rațu, Iuliana Motrescu, Irina Gabriela Cara, Manuela Filip, Denis Țopa, Gerard Jităreanu

**Affiliations:** 1Department of Pedotechnics, Faculty of Agriculture, “Ion Ionescu de la Brad” University of Life Sciences, 3 Mihail Sadoveanu Alley, 700489 Iasi, Romania; florina.stoica@iuls.ro (F.S.); denis.topa@iuls.ro (D.Ț.); gerard.jitareanu@iuls.ro (G.J.); 2Department of Food Technologies, Faculty of Agriculture, “Ion Ionescu de la Brad” University of Life Sciences, 3 Mihail Sadoveanu Alley, 700489 Iasi, Romania; 3Department of Exact Sciences, Faculty of Horticulture, “Ion Ionescu de la Brad” University of Life Sciences, 3 Mihail Sadoveanu Alley, 700489 Iasi, Romania; iuliana.motrescu@iuls.ro; 4Research Institute for Agriculture and Environment, “Ion Ionescu de la Brad” University of Life Sciences, 700490 Iasi, Romania; filipmanuela@yahoo.com

**Keywords:** black carrot pomace, anthocyanins, antioxidant activity, natural ingredients, value-added products

## Abstract

Researchers and food manufacturers are investigating the use of fruit and vegetable by-products as nutrient-dense food ingredients in response to increasing consumer requests for healthier and more natural foods. Black carrot (*Daucus carota* L.), a root vegetable variety of deep purple carrot, is a valuable source of nutrients with excellent health benefits and nutraceutical effects. Black carrot pomace (BCP), a by-product of industrial juice extraction, is abundant in bioactive compounds, dietary fiber, antioxidants, and pigments such as anthocyanins. Value addition and sustainability are perspectives provided by using this underutilized agricultural by-product in food applications. With an emphasis on BCP powder’s effects on phytochemical and physicochemical qualities, mineral and color characteristics, and sensory aspects, this study aims to assess the effects of adding BCP powder to yogurt formulations. The findings show that the addition of BCP powder improved the nutritional, and the color of the yogurts, providing a visually appealing product. Moreover, adding the BCP powder raised the amount of phytochemicals and the antioxidant activity in the final product’s formulation. The manufacturing of such products can not only aid in promoting sustainable food production but also offer consumers a wider range of innovative food options with improved properties.

## 1. Introduction

Consumers’ changing habits have led to a notable rise in the demand for functional meals, which are seen as wholesome. The attention to novel antioxidants derived from natural sources has been recently increasing. Due to their abundance in several categories of polyphenols, fruits and vegetables are receiving a lot of attention [1]. Farmers all over the world must cultivate a greater variety of crops. They can do this by using fertilizer and irrigation to make current farmlands more productive, or they can try new methods like precision farming [2,3].

Carrots (*Daucus carota* L.) are one of the most consumed vegetables and a significant root crop farmed worldwide due to their high nutritional content. Carotenoids from *Daucus carota* ssp. *sativus* and anthocyanins from *Daucus carota* L. ssp. *sativus* var. *atrorubens Alef*. are the two groups of colored substances that can be found in carrots [4]. With an annual growth rate of around 10.5%, the world’s carrot production reached 40 million tons in 2015 and 44.80 million tons in 2019 [5]. The growing importance of carrot processing and production into various commodities, along with inadequate infrastructure and management techniques, has resulted in the generation of enormous quantities of waste. In 2019, the worldwide production of carrot pomace amounted to approximately 45 million tonnes. Carrot residues, which comprise 30–50% of the pomace and 15% of the peels of the raw material after industrial processing, contain an abundance of bioactive compounds [6]. These byproducts are primarily utilized as fertilizer and animal feed, or they are disposed of in landfills where they negatively impact the environment. However, substantial quantities of residual valuable compounds, which have been associated with a variety of nutritional and health benefits, are present in this waste (minerals, fibers, anthocyanins, polyphenols) [7].

Though black carrots are predominantly grown and consumed in oriental nations like India, Afghanistan, Turkey, Egypt, Pakistan, and the Middle and Far East, it is thought that they are considerably older than the more widely recognized orange carrot variety [8]. It is primarily utilized in the pigment industry to produce an anthocyanin-rich concentrate. Black carrots’ intense purple color is attributed to anthocyanin compounds. This root vegetable contains minerals (Ca, P, Mg, and Fe), proteins, carbohydrates, dietary fiber, vitamins A, B1, B2, B6, B9, C, E, and K, and bioactive substances. Because bioactive components are present, it has significant health-promoting qualities and substantial antioxidant activity [9,10].

Despite its evident benefits as a natural food coloring, a natural antioxidant, and a vegetable with a high mineral content, black carrots are very underutilized and are not favored by consumers. Not enough studies have been performed to support the use of black carrots in the food industry, despite their many health benefits. Turkey produced approximately 14,000 tonnes of black carrot concentrate (approximately 60% dried matter) in 2013, the majority of which was exported as a natural food coloring agent to countries including Italy, Denmark, France, Japan, and China. The anticipated continuation and perhaps acceleration of this trend in the coming years can be attributed to the increasing popularity of black carrot juice and concentrate. This massive processing generates enormous quantities of pomace, the dispersal of which is an overwhelming challenge. Juice extracts high levels of anthocyanin and polyphenols, but black carrot pomace retains a large number of residual high-bioactivity polyphenolic chemicals that can be used in a variety of industries, including the food and nutraceutical sectors [11].

The anthocyanin pigments, which are constituents of the flavonoid group and have significant antioxidant potential, are responsible for the dark purplish-black hue of the black carrot. Compared to orange-colored carrots, black carrots have about 2.5 times the antioxidant potential. The most prevalent anthocyanin found in black carrots is acylated cyanidin-3-(p-coumaroyl)-diglucoside-5-glucoside [12]. Anthocyanin acylation increases anthocyanins’ resistance to changes in pH, heat treatments, light exposure, and oxygen. In the food sector, natural sources of anthocyanins are frequently utilized as a good substitute for artificial colorants. Because anthocyanins are water soluble, they are a desirable and natural food coloring for a variety of uses. Similarly, because of their remarkable pigmentation and antioxidant qualities, they have a variety of uses in pharmaceuticals, cosmetics, and health products [13]. Antiproliferative, anti-inflammatory, antioxidant, antiviral, antiallergic, stomachic, and bactericidal properties are among the health benefits associated with these pigments [9]. The anthocyanin pigment found in black carrots has both structural and functional qualities that allow for a wide range of food applications, particularly in confections and low-acidity beverages. Moreover, anthocyanin derived from black carrots is used to color a wide range of culinary items, including fruit juice concentrate, nectars, soft drinks, jellies, yogurt, and marmalades [4,14]. Apart from their remarkable hue, black carrots rich in anthocyanins have numerous health advantages. Being potent antioxidants, they may fend off heart disease, some types of cancer, diabetes, and obesity, and even enhance cognitive function [15].

Concerns about the safety of synthetic food dyes have led to demand among stakeholders for natural colorants. Approved artificial coloring substances are being examined for possible toxicity as frequent exposure to increasing concentrations of these substances may cause mutagenesis and cancer. For many years, the benefits of fermented milk products on human nutrition and health have been widely acknowledged, and these products are consumed worldwide [16,17]. Yogurt is the most appreciated fermented dairy product due to its distinct nutritional and sensory qualities. Yogurts have undergone innovation to improve their organoleptic, nutritional, and health benefits. Some of these innovations include the addition of other non-dairy components (prebiotics, synbiotics, minerals, vitamins), protein enrichment, dietary fibers, and phytochemicals. Researchers employed anthocyanin-rich ingredients to color yogurt, including mulberry juice [18], dried roselle calyx [19], and powdered Peruvian berries [20]. They noticed improvements in the yogurt’s color as well as its bioactivity. Fruit-fortified dairy products are already on the market and are highly accepted by consumers. Dairy products can be effectively used as food carriers for natural colorants derived from black carrot polyphenols and anthocyanins.

In this research, different concentrations of BCP powder were incorporated into yogurt during its production process, and the impact on its physicochemical, phytochemical, antioxidant activity, color, sensory, and nutritional properties was evaluated. This study also investigates the stability of bioactive compounds and color during yogurt processing and storage. The present study underlines the potential of BCP powder as an innovative and sustainable ingredient in the dairy sector, offering the dual benefit of reducing waste and enhancing the nutritional profile of yogurts.

## 2. Materials and Methods

### 2.1. Reagents and Chemicals

Folin–Ciocalteu reagent, 1% citric acid, ethanol, 6-hydroxy2,5,7,8 tetramethylchromane-2-carboxylic acid (Trolox), 2,2-diphenyl-1-picrylhydrazyl (DPPH), sodium hydroxide, sodium nitrite, Gallic acid, potassium chloride solution, sodium acetate solution, aluminum chloride, and sodium carbonate were obtained from Sigma Aldrich Steinheim (Darmstadt, Germany). Analytical-grade reagents were the only ones utilized in the experiments. 

### 2.2. BCP Powder Preparation

Black carrots (*Daucus carota* L. ssp. *sativus* var. *atrorubens Alef*.) were purchased from a supermarket (Iasi, Romania) in June 2023. The raw materials were sorted, washed with distilled water, shredded beforehand, and subjected to squeeze. The carrot juice was extracted using a Bosch MES3500 (Philips Consumer Lifestyle B.V., Drachten, The Netherlands) device, yielding the black carrot pomace.

The black carrot pomace was stored in plastic bags at a temperature of −20 °C before the freeze-drying experiment was conducted. The black carrot pomace was freeze-dried for 48 h at −42 °C under a pressure of 0.10 mBar up to a moisture content (MF-50 moisture analyzer, A&D Company, Tokyo, Japan) of 9.0%, in a freeze-dryer (BIOBASE BK-FD10T equipment, Jinan, China), with a 0.5-cm layer thickness. The resultant BCP was pulverized (mean particle diameter of 1 mm) for 30 s in a grinding mill, kept at room temperature in an airtight glass jar, and utilized for further extraction. Prior to the supplementation of the yogurt, the BCP powder underwent decontamination through sterilization using a UV lamp.

### 2.3. Extraction of Phytochemicals from BCP Powder

For the ultrasound-assisted extraction of bioactive compounds from BCP powder, the method described by Lee et al. [21] was used, with a few minor adjustments. A total of 1 g of BCP powder was solubilized with 10 mL of 70% ethanol acidified with 1% citric acid solution, in a ratio of 1 to 9, acid/solvent. The mixtures were treated with ultrasound using a sonication bath (Elmasonic S 180 H, Elma, Singen, Germany) for 40 min at 40 kHz, power of 100 W at 25 °C, and then centrifuged for 10 min at 6000 rpm and 4 °C. The supernatant was then used for phytochemical characterization experiments.

### 2.4. Extract Characterization

The BCP extract was characterized in terms of total monomeric anthocyanin contents, total flavonoids, total polyphenols contents, and DPPH radical scavenging activity.

#### 2.4.1. Determination of Total Anthocyanin Content

A pH differential approach was employed to determine the total monomeric anthocyanins content of *BCP* extract according to the modified protocol used by Lee et al. [21]. The samples were diluted (D = 1:10) before analysis. A total of 200 µL of extract and 800 µL of a buffer solution with a pH = 1.0 (potassium chloride, 0.025 M) or pH = 4.5 (sodium acetate, 0.4 M) were employed. The absorbance of the diluted extracts was measured at two different wavelengths: 520 nm and 700 nm, after a 15 min rest in the dark, using a UV-Vis spectrophotometer (Analytik Jena Specord 210 Plus, Jena, Germany).

The findings were presented as milligrams cyanidin-3-glucoside (C3G) per gram of dry weight (dw). The total anthocyanin content was calculated as follows (1):(1)$$Total\;Anthocyanins\;(mg\;C3G/g\;dw)\;=\;\frac{A\;\times \;MW\;\times \;DF}{\varepsilon \;\times \;l\;\times \;m}$$
where A = (A520–A700 nm) pH 1.0—(A520 nm–A700 nm) pH 4.5; MW is molecular weight, 449.2 g/mol for cyanidin-3-glucoside; l is cuvette pathlength, 1 cm; DF is the dilution factor; m is the amount of sample; and ε is molar extinction coefficient for cyanidin-3-glucoside, 26,900 L/mol/cm.

#### 2.4.2. Determination of Total Polyphenol Content

To determine the total polyphenol content, the Folin–Ciocâlteu colorimetric method was used, as described by Dewanto et al. [22]. Briefly, 200 μL of extract, 15.8 mL of distilled water, and 1 mL of the reagent Folin–Ciocalteu were added. After 10 min, 3 mL of 20% sodium carbonate solution was added and the mixture was kept for 60 min at room temperature in the dark. Then, the absorbance of the mixture was evaluated at 765 nm. Additionally, a standard curve was created utilizing different Gallic acid concentrations, and the results were reported in terms of mg Gallic acid equivalents (GAEs) per g dry weight (dw).

#### 2.4.3. Determination of Total Flavonoid Content

The method used by Dewanto et al. [22] was used to determine the total flavonoid content of the supernatant from the BCP extract, with some modifications. Briefly, a mixture containing BCP extract (250 µL) with ultrapure water (1250 µL) and 5% NaNO_2_ solution (75 µL) was allowed to react for 5 min. Subsequently, 150 μL 10% aluminium chloride solution, 500 μL NaOH solution 1 M, and 3000 µL ultrapure water were added. The mixture absorbance was immediately evaluated at a wavelength of 510 nm. The total flavonoid content was assessed by plotting the catechin calibration curve and the findings were expressed as mg catechin equivalents (CEs) per g dry weight (dw).

#### 2.4.4. Determination of the Antioxidant Activity

The antioxidant activity was expressed by the BCP extract’s capacity to scavenge the DPPH (2,2-diphenyl-1-picrylhydrazyl) radical according to Castro-Vargas et al. [23]. Briefly, 3.9 mL of methanolic DPPH solution (0.1 M) was combined with volumes of 0.1 mL extract, and the mixture was left to react for 30 min at room temperature in the dark. For the blank sample, −0.1 mL of methanol was mixed with 3.9 mL of DPPH solution. Using a UV–visible spectrophotometer (Analytik Jena Specord 210 Plus, Jena, Germany), the absorbance was measured at a wavelength of 515 nm. Additionally, a standard curve was created using different Trolox concentrations. The findings were expressed as µmol of Trolox equivalent (TE) for each g of dry weight (dw). The radical scavenging activity was expressed also as the percentage of inhibition based on Equation (2):(2)$$\mathrm{DPPH}\;\mathrm{scavenging}\;\mathrm{activity}\;(\%)\;=\;\frac{Abs\;Control\;-\;Abs\;Sample}{Abs\;Control}\;\times \;100$$ where Abs Sample is the absorbance value of the DPPH solution combined with BCP extract; Abs Control is the absorbance value of the DPPH solution only.

### 2.5. Preparation and Characterization of Supplemented Yoghurt

To obtain the enriched yogurt and test its functionality, the BCP powder was used in two different ratios (1%-YBCP1 and 2%-YBCP2). The control sample (YC) consisted of yogurt without added powder.

The technological step of preparing yogurt with black carrot pomace started with the raw material and ingredient receipt stage (cow’s milk, lactic cultures (Streptococcus thermophilus and Lactobacillus delbruckii subsp. bulgaricus, YF-L812 commercial product, Chr. HANSEN, Hørsholm, Denmark), BCP), along with the addition of auxiliary components (Figure 1). The process for keeping auxiliary and raw materials was as follows: milk was stored in temperature-controlled isothermal tanks (0–4 °C); yogurt lactic cultures were maintained at −18 °C with temperature and humidity monitoring; bioactive powder (BCP 1 and 2%) and packaging materials were stored in areas with adequate ventilation, natural light, and no external odors at room temperature. To guarantee product compliance, air quality parameters such as temperature and relative humidity were also tracked. Following filtration, the milk was pasteurized for ten minutes at 90 °C. The milk was then allowed to cool while the temperature was kept under observation, maintaining it constant at 42 °C.

The next stage of the technological process involved dosing and preparing the lactic starter cultures, starting with the standard 50 U culture per 250 L of milk, per the manufacturer’s guidelines. Three batches (each containing 35 L of milk) were separated. The initial portion, devoid of BCP inclusion in the mixture, was designated as the control batch (YC—control batch). Measures of 1% BCP (YBCP1, yogurt with 1% BCP) and 2% BCP (YBCP2, yogurt with 2% BCP) were added to the milk in the second and final portions, respectively (Figure 1). Furthermore, the number of lactic starter cultures (mg/batch) in every batch was monitored. The bioactive component (BCP) was introduced as the following technological step, and its level (1% and 2%/batch) was examined. The yogurt was distributed into 120 mL PET glasses, then thermo-welded at 227 °C. There was a record of the quantity of glasses. After that, the mixture was placed in plastic cups and incubated at 43 °C using an incubator (IT 40 thermostatic chamber, made by Electronic April S.R.L., Cluj-Napoca, Romania) for 360 min, until a firm coagulum and a pH range from 4.3 to 4.5 were achieved (5–6 H). After 1, and 21, days of manufacturing, the three duplicates of the stirred yoghurt were analyzed, and the yoghurts were kept at 4–6 °C for storage.

### 2.6. Characterization of Physicochemical and Phytochemicals of Supplemented Yogurt

The physicochemical properties of the supplemented yogurt samples, including their moisture and dry matter contents, pH, fat, carbohydrates, protein, fiber, ash, and energetic values were evaluated using AOAC procedures [24,25].

The total anthocyanins, flavonoids, polyphenolic content, and antioxidant activity of supplemented yogurt enhanced with BCP powder were estimated using the procedures described above in Section 2.4.

Syneresis of the yogurt samples was measured according to the method used by Wijesinghe et al. [26], with slight modification. In summary, 10 g of every yogurt sample was separately put on a filter paper sheet and allowed to settle on top of a funnel. After 10 min of vacuum draining, the remaining yogurt was weighed.

Syneresis was calculated as follows:(3)$$\mathrm{Free}\;\mathrm{whey}\;(\%)\;=\;\frac{mass\;of\;initial\;sample\;(g)\;-\;mass\;of\;sample\;after\;filtration\;(g)\;\times \;100}{weight\;of\;initial\;sample\;(g)}$$

### 2.7. Measurement of Minerals

The mineral contents of the tested samples were determined by atomic absorption spectrometry (ContrAA 700, Analytik Jena, Jena, Germany), with a flame atomizer system, and values were expressed in mg/100 g dw.

Measures of the mineral contents (Ca, P, K, Mg, Zn, Fe, and Na) of the raw plant material and yogurt samples were carried out using a MiniWAVE Microwave (SCP Science, Baie-d’Urfé, QC, Canada) digestion system equipped with a 50 mL Teflon vessel. A total of 1 g of the homogenized sample was weighed into a Teflon vessel and digested using a nitric (HNO_3_) and hydrochloric (HCl) acid mixture (8:1). Digestion was performed under the following conditions: temperature—180 °C; digestion time—20 min; microwave power—1000 W. After the sample had cooled, it was carefully transferred into a 25 mL volumetric flask and diluted with ultrapure water until the mark. A blank sample was added in every digestion run, and each sample was prepared in triplicate.

### 2.8. Storage Stability of the Phytochemical Compounds

The yoghurt samples were kept in light-resistant PET glass containers at 5 °C, in the dark. They were analyzed for their bioactive contents (anthocyanins, flavonoids, and polyphenols) and antioxidant activities, as previously described, during 21 days of storage.

### 2.9. Color Evaluation of Supplemented Yoghurt Samples

The color of the BCP-enriched and control yogurt was determined for CIE (Commission Internationale de l’éclairage) L*, a*, b*, Chroma, color intensity, and hue angle value using a MINOLTA Chroma Meter model CR-410 (Konica Minolta, Osaka, Japan).

Whiteness, redness to greenness, and yellowness to blueness were represented by the color coordinates L*, a*, and b* as reflected in CIELAB, respectively. Before determining color, the product samples (in triplicate) were brought to room temperature. Using the formulas provided below, the primary data were calculated for hue angle, color intensity, and Chroma.

Hue angle = 360 + arctan(b*/a*) for quadrant IV (+a*, −b*), which describes visual color appearance.

Chroma = $\sqrt{{{(a}^{*})}^{2}+{{(b}^{*})}^{2}}$, which describes color intensity.

ΔE = $\sqrt{{L^{*2}+a}^{*2}+b^{*2},}$ the total color difference [27].

### 2.10. Scanning Electron Microscopy Analysis

A scanning electron microscope (SEM) (Quanta 450, FEI, Thermo Fisher Scientific, Hillsboro, OR, USA) and an energy dispersive X-ray detector (EDS) (EDAX, AMETEK Inc., Berwyn, PA, USA) were used to characterize the morphology of the samples and detect their elemental composition. A small quantity of each sample was pasted in an approximately 2 mm layer onto aluminum stubs, and then placed in the analyzing chamber. The TEAM version V4.1 system, created by EDAX Inc., performed the EDS spectra analysis. Before the investigation, a standard AlCu sample—a copper foil set on an aluminum grid—was used for calibration. The samples were examined under low vacuum conditions, with a pressure of roughly 6.1 × 10^−4^ Pa, an electron acceleration voltage of 15 kV, and a 500× (10 μm) magnification.

### 2.11. Sensory Evaluation of Supplemented Yoghurt Samples

The sensorial properties of the yoghurt samples were assessed by 20 panelists from the Faculty of Agriculture, University of Life Sciences, staff. All of these panelists, who were highly skilled in food technology, were selected based on their willingness to take part and their expertise in dairy products. The individuals were regular consumers of yogurt and did not exhibit any allergic reactions to it. In addition, the panelists underwent two training sessions to examine and deliberate on the several sensory descriptors that were assessed, such as the modifications in color, texture, and flavor of the yogurt samples. The appearance, specific color, flavor, texture, taste, odor, aftertaste, and acceptability were evaluated on a nine-point scale (1—extremely dislike, 9—extremely like) [28]. The samples were delivered for analysis in clear, colorless plastic containers, each of which had a three-digit code on it. The panelists were given water and bread to cleanse their palates in between testing samples. Additionally, they were invited to document any critiques they had regarding the evaluated products.

### 2.12. Statistical Analysis

The findings were expressed as the mean ± standard deviation (SD) for every sample. The statistical differences amongst the samples were assessed using Minitab 19.0 (free trial). When significant differences were found in this regard, one-way analysis of variance (ANOVA) and Tukey’s test with a 95% confidence interval were used.

## 3. Results and Discussion

### 3.1. Phytochemical Characterization of BCP Powder

Among the most frequently underutilized vegetables on a daily basis are black carrots. They have a large amount of bioactive compounds (anthocyanins, flavonoids, and polyphenols) and antioxidants. Table 1 presents the phytochemical content and color properties of BCP powder. To extract biologically active chemicals from BCP using the ultrasound-assisted extraction method, 70% ethanol that had been acidified with 1% citric acid was selected.

The content of anthocyanins in the BCP extract was 3.46 ± 0.34 mg C3G/g dw and the DPPH scavenging activity was 24.19 ± 0.66 µmol TE/g dw. Additionally, previous studies exhibited that BCP is a rich source of anthocyanin components such as cyanidin-based anthocyanin with different sugar moieties acylated or non-acylated [11,15,29]. The total polyphenols and flavonoid contents were found to be 22.89 ± 0.87 mg GAE/g dw and 18.49 ± 0.86 mg QE/g dw, respectively. The BCP powder’s color changes were as follows: the L* value was 21.29, the a* value was 8.17, and the b* value was −1.01. Based on color characteristics, values for chroma, hue angle, and total color change (ΔE) were estimated at 8.23, 359.88, and 22.82. The powder was located in quadrant IV (+a*, −b*), according to the results of the color indices. The high concentration of biologically active compounds in BCP powder found in our study is consistent with previous reports by Polat et al. [4], who reported an anthocyanin content of 101.8 ± 0.24 mg/100 g dw and a phenolic content of 81.6 ± 0.4 mg/kg dw in dried black carrot pomace. In their study, Algarra et al. [15] reported lower values of bioactives in black carrot extracts compared to our findings, such as total phenols of 492.0 ± 63 mg GAE/100 g fresh weight, anthocyanins of 126.4 ± 6.0 mg/100 g fresh weight, and a reducing capacity by DPPH method of 240.0 ± 54.0 µM TE/100 g fresh weight.

The estimated minerals in BCP powder suggested that black carrot by-products could also be considered a good source of calcium, sodium, magnesium, potassium, and phosphorus [30,31]. Nevertheless, additional research is necessary to evaluate their bioavailability. Encouraging the use of native vegetables in diets could potentially reduce food insecurity and malnutrition in developing nations [32].

The composition of black carrots by-products varies depending on the variety, the agronomic conditions of the region in which they were grown, the extraction techniques used (e.g., kind of solvent, temperature, pH, and light intensity), and the assessment techniques used. Meanwhile, research has shown that BCP extracts are an excellent source of phytochemicals with antioxidant activity and can be added to food as an ingredient to help lower the amount of agro-industrial residues.

### 3.2. Characterization of Bioactive Potential of Supplemented Yoghurts and Storage Stability of the Samples

Black carrots are mostly composed of polyphenols and anthocyanins, which are widely recognized for their potent antioxidant action and ability to impart color [9,33]. The phytochemical profile and antioxidant activity of control and experimental yoghurts are reported in Table 2.

The experimental dairy products had anthocyanin contents ranging from 9.15 ± 0.24 to 18.06 ± 0.42 mg C3G/100 g. Plenty of research has been performed recently on the application of polyphenolic chemicals, like anthocyanins, and their potential to suppress a range of degenerative and chronic illnesses [34]. Black carrot’s by-products’ primary bioactive components are anthocyanins, flavonoids, and polyphenols, which are well-known for their potent antioxidant action and ability to color food [6]. The addition of the BCP powder into the yogurts’ formulation resulted in increases in anthocyanin, flavonoid, and polyphenol levels. Additionally, incorporating black carrot concentrate into dairy products (ice cream, yogurt, and buttermilk) resulted in a high amount of anthocyanins and flavonoids and a high Folin–Ciocalteu reducing capacity [35]. The experimental samples showed good bioactive contents, while the control samples showed negligible quantities. This suggests that the dairy products included trace levels of polyphenols, which may have come from milk. Polyphenolic compounds can be found in milk from animal feed, amino acid catabolism, and/or from the environment, according to O’Connell and Fox [36]. As a result of the Maillard reaction, polyphenol levels in milk may also rise during heating.

The samples’ total anthocyanin, total polyphenol, and total flavonoid contents declined over time (*p* < 0.05). Additionally, DPPH scavenging activity decreased in all samples after 21 days, while YBCP1 (1%) and YBCP2 (2%) activity stayed higher than in the control. Arts et al. [37]’s research suggests that the binding interaction between dairy protein and polyphenol is the cause of this decline. Furthermore, the explanations provided by Dubeau, Samson, and Tajmir-Riahi [38], and Ersöz, Kınık, Yerlikaya, and Açu [39], supported our findings and demonstrated that the reduction in antioxidant activity caused by protein–polyphenol binding is achieved by lowering the amount of free hydroxyl radicals.

The results shown in Table 2 demonstrate that adding BCP powder to yogurts improves their quality, as seen by the satisfactory concentrations of anthocyanins and polyphenols. These natural ingredients help to develop dairy products with satisfactory antioxidant activity.

### 3.3. Physico-Chemical Characterization of Supplemented Yogurt Samples

Yogurt is probably one of the most well-known and consumed dairy products worldwide. The physicochemical parameters of control and experimental yogurts after obtaining the samples and after 21 days of refrigerated storage are reported in Table 3. The nutritional composition of yogurts will change when natural functional components, such as those derived from vegetables/fruits, grains, and other meals, are incorporated [40].

The addition of BCP powder to yogurt substantially improved its chemical composition in comparison to the control. The proximate composition of the yogurt with BCP and without BCP showed a significant difference (*p* < 0.05) between samples. The addition of BCP causes significant variations (*p* < 0.05) in moisture, fat, carbohydrates, protein, fiber, ash, energetic value, and residual dry matter content.

In addition, the enrichment of yoghurt with BCP induced a carbohydrates and moisture decrease, which was followed by a protein, fat, fiber, ash and total solids increase with the increase amounts of added-BCP. A considerable amount of dietary fiber and crude protein in black carrot powder could be responsible for these findings [9,41]. With an increased proportion of added BCP powder, the total solid and total fat contents in yoghurt grew significantly (*p* < 0.05). During the 21 days of storage, differences between samples (*p* < 0.05) were noticed only in the case of YBCP1 regarding fat content. In contrast, in terms of the total solids, the differences were significant (*p* < 0.05) between the enriched samples (YBCP1, YBCP2). Moreover, YC and YBCP1 recorded a higher carbohydrate content compared to that identified in YBCP2. During the storage period, the carbohydrate content increased significantly (*p* < 0.05) between enriched batches.

The protein level data show a statistically significant difference (*p* < 0.05) between the YBCP2 yoghurt (5.11 ± 0.03%) compared to the value obtained for YC (4.30 ± 0.18%). This difference is influenced by the content of BCP added to each product. Significant changes in protein levels were observed during storage in the supplemented yogurts (*p* < 0.05). The decrease in the protein content of the yogurt with the addition of BCP powder could be attributed to several factors (the syneresis rate, protein–polyphenol complexes which may be more susceptible to proteolytic enzyme activity) related to the biochemical and interactive effects between the components of the yogurt and the added BCP powder [42,43].

The results displayed that the moisture was lower as the BCP was added, and the ash content was higher as the enhancement of yogurt was realized. The yogurts with the highest ash concentration were found in YBCP2 (1.71 ± 0.03%), whereas the lowest ash content was observed in YC (0.57 ± 0.02%). These differences were statistically significant (*p* < 0.05). During storage, no significant differences (*p* > 0.05) were established between the enriched samples regarding ash.

The data on fiber content show significant differences (*p* < 0.05) among the three analyzed batches produced in the first phase. Specifically, there were no fiber identifications at YC, but there were significant differences between YBCP1 (1.69 ± 0.02%) and YBCP2 (2.11 ± 0.01%) on the initial day (*p* < 0.05). No significant variations were found during storage (*p* > 0.05). The differences in the energy value at the initial control were statistically significant (*p* < 0.05). The average energy value for YC was 75.09 ± 0.33 kcal/100 g, whereas for YBCP2 it was 73.53 ± 0.06 kcal /100 g. The findings of this investigation were consistent with those reported in studies carried out with yogurts supplemented with grape pomace and grape extract [44,45].

The inclusion of BCP resulted in a statistically significant (*p* < 0.05) reduction in the pH of the samples to which BCP was added. The pH value at the initial moment for YBCP1 was measured to be 4.43 ± 0.01, while for YBCP2 it was 4.39 ± 0.01. Throughout the storage period, the pH value displayed a decline in all samples. Yogurt’s excessive acidification refers to the decrease in pH that happens when yogurt undergoes storage [46].

In simple terms, the findings of the proximate chemical composition analysis indicated that the addition of powdered bioactive ingredients during yogurt manufacturing resulted in noticeable variations. The yogurt samples’ nutritional composition increased with BCP content, indicating a greater nutritional quality compared to the control sample.

Table 3 depicts the syneresis results obtained from the yogurt samples made with varying concentrations of BCP. Syneresis is the primary flaw of yogurt. It impacts both the structural and textural characteristics of yogurt during storage. A yogurt sample with a lesser degree of syneresis value will be more desirable to consumers. The syneresis values exhibited significant differences (*p* < 0.05) both among the samples and over the storage days. The addition of BCP reduced the syneresis of the yogurt compared to the control; on the 21st day, the control showed the highest values, while on the first day, the sample that included 2% BCP showed the lowest values. The rise in dry matter content can be the cause of the decrease in syneresis on day one. Throughout storage, all yogurts showed controlled growth, according to the syneresis. Mahdian and Tehrani [47] showed that, when the overall solid content rose, there was a discernible decrease in yogurt syneresis. Therefore, between day 14 and day 21 of storage, there was a rise in syneresis, which is the samples’ capacity to hold water. Dönmez et al. [48] stated that reducing syneresis through BCP increases the water content retained inside the gel network, resulting in decreased serum release in yogurt. Ghadge et al. [49] found that the acidity of yogurt increased due to the rise in syneresis during storage.

### 3.4. Mineral Profile

Yogurts’ macro- and micro-element contents are tightly connected to the original milk that was utilized to make those products. During fermentation, the mineral content of milk and yogurt remains unchanged [50]. A food high in nutrients, yogurt is an excellent option for acquiring certain minerals including calcium, magnesium, phosphorus, and other elements. These necessary micronutrients improve human health, particularly by lowering the risk of illness. Many health conditions, including mental retardation, learning difficulties, blindness, low employment efficiency, and early death, are brought on by their deficiency [51,52].

Table 4 displays the mineral contents analysis for the prevailed samples from yogurt without BCP (control sample) and supplemented yogurt with BCP used in two different ratios (1% and 2%).

The amounts of mineral contents increased in congruence with the increased amount of BCP powder. The differences between the yogurt samples in terms of the amounts of Na, Mg, P, K, Ca, Fe, and Zn were found to be statistically significant (*p* < 0.05). Yogurt samples contain additional essential elements including calcium and phosphorus. Notably, the highest concentration of Ca was discovered in the YBCP2 (195.25 ± 0.34 mg/100 g) sample with 2% BCP powder. The results of the statistical analysis showed that there were statistically significant differences in the amount of P in each yogurt sample. The maximum amount of P was found for the yogurt with 2% BCP powder, followed by the yogurt with 1% BCP powder. The control sample showed the lowest P amount (54.08 ± 0.64 mg/100 g).

Additionally, the yogurt enriched with 2% BCP powder had the highest amounts of K and Mg, followed by the yogurt containing 1% BCP powder while the lowest amounts were found in the control sample. Regarding the content of zinc, iron, and sodium, there were significant differences (*p* < 0.05) between the samples with the addition of BCP powder and the control sample. The incorporation of 1% and 2% of BCP (YBCP1 and YBCP2) led to a higher content of Zn, Fe, and Na as compared to plain yogurt. The study’s findings showed that adding powdered BCP raw material to yogurt increased the quantity of minerals in the final product from 0.78 to 30.07 times. As can be seen in Table 1, freeze-dried black carrot powder is rich in Ca, K, Mg, P, and Na. Hence, BCP powder is suitable for incorporation as a bioactive ingredient in foods.

Comparing the mineral analysis results of the yogurt samples with those reported by Çardak et al. [53], the amounts of minerals like Ca, K, Na, and Fe appeared to be higher in their study, whereas those of Mg and Zn were lower. Additionally, it was observed that adding concentrated freeze-dried black carrot fiber at different ratios (0%, 0.25%, 0.5%, and 1%) to an ayran-containing black carrot fiber enhanced its Ca, K, Mg, Na, P, and Zn levels significantly [54]. Other authors such as Kulaitienė et al. [55] reported that the incorporation of 1% of different powders (beetroot, mulberry leaves, nettle leaves, and rosehip fruit) into yogurt bites allowed for a significant increase in the amounts of the investigated minerals (Ca, K, P, Mg, Fe, and Zn).

### 3.5. Color Evaluation of Supplemented Yoghurts

The color of fermented products is one of the primary characteristics that influence consumers’ perceptions of quality and product preferences; this feature is just as significant as their shelf life [56]. The results of the color measurements (L*, a*, b*) of the yogurt samples on the first control day and after 21 days of storage at 4 °C are presented in Table 5.

The L* values, which show the whiteness of the product, fluctuated between 58.62 ± 0.13 and 48.23 ± 0.93 for the YBCP1 and YBCP2 samples. The color of the yogurts had been substantially impacted by the addition of BCP, according to the examination of the color parameters. To be more precise, the experimental yogurts had the lowest lightness (L*) and blueness (b*) values and the maximum redness (a*) values. Statistically significant changes were seen in the L*, a*, and b* values of yogurt samples in 21 days of storage (*p* < 0.05). The YBCP2 sample’s total color change characteristic, or ΔE, ranges from 58.20 to 57.87 after being stored for 21 days. When BCP powder was added, there was a small decrease in ΔE during storage. The color’s Chroma, which shows how intense and saturated it is, was greatest in the YBCP2 yogurt. Since the hue angles were around 360°, the value of the hue angle was related to the color that was received and showed that both samples were red. Most of the time, an angle of 0° or 360° stands for red, while an angle of 90°, 180°, or 270° stands for yellow, green, or blue [57].

The color of the supplemented yogurts was darker when black carrot powder was added in an increasing ratio (*p* < 0.05). As the amount of black carrot rose, the color of the supplemented yogurts approached zero, but the control yogurt with the lightest color had the highest L* value. Baria et al. [58] noted the presence of a decrease in the L* value of yogurts when the ratio of black carrot concentrate (0.5%, 1%, 1.5%, and 2%) increased. Madora et al. [43] attributed the decrease to the increased concentration of carrot powder (1%, 2%, and 3%).

The a* values of the yogurt samples changed significantly (*p* < 0.05) when black carrot powder was added. YBCP2 having 2% black carrot powder appeared to be closer to the color red than the control sample, which was perceived as being closer to the color green. The predominant hue of the yogurt containing black carrots was red, which was caused by the conversion of anthocyanins at low pH values. Similar to the values in this study, several researchers have found higher a* values in yogurts with added carrot powder, carrot juice, or carrot concentrate than those in the control sample [43,58,59].

It was found that the control yogurt’s b* values were fairly near to the hue yellow. The hue of the yogurt significantly approximated blue after the addition of black carrot powder (*p* < 0.05). Sample YBCP2 showed greater b* values on day 21 of storage, compared to sample YBCP1 which showed lower values. Additionally, Baria et al. [58] observed that, when the concentration of black carrot increased, b* values decreased.

### 3.6. Microstructure Analysis

Figure 2 displays the scanning electron microscopy (SEM) pictures of the yogurt samples. The yogurts incorporated with BCP powder had a distinct microstructure from the control yogurt (Figure 2a). There were many gaps and pores visible, the control’s surface was uneven, and the protein connection was crooked. When BCP was added, the yogurt samples’ pore spaces and channels with denser or more homogeneous protein aggregation and microstructure were greatly reduced. BCP particles could be present in the empty spaces around the casein aggregates. They increased the aggregates’ ability to hold water and decreased whey separation by attaching to the serum channels around them.

The control yogurt sample exhibited denser chains, a reduced number and size of empty spaces, and smaller clusters of casein micelles. In contrast, the yogurt samples enriched with BCP displayed a distinct pattern. Adding BCP to yogurt resulted in a more uniform and evenly distributed casein network, which had a smoother texture and minimized gaps. The overall structure of the yogurt, however, appeared slightly coarser. This phenomenon can be attributed to the linkage between BCP and milk proteins, facilitated by hydrocolloids and the stability of the emulsion. The results correspond with the observations of Ibrahim and Khalifa [60]. The BCP-supplemented yogurt exhibited reduced variability in the appearance of casein micelles. The variations observed were likely attributed to the formation of casein BCP complexes, primarily facilitated by hydrophobic interactions between casein micelles and BCP, as suggested by Brodziak et al. [61]. The dietary fiber of BCP has the ability to bind water and raise the water-holding capacity of yogurt samples, which improves protein network connections [62].

### 3.7. Sensorial Analysis of Supplemented Yoghurts

Sensory evaluation of the supplemented yogurts was realized using a nine-point hedonic scale. The average scores obtained from the sensory evaluation are displayed in Figure 3. The attributes followed were appearance, color, flavor, taste, texture, odor, aftertaste, and acceptability.

The hedonic test was used to assess the degree of liking of yogurts developed by using different levels of BCP. Yogurts containing 1% and 2% BCP powder exhibited higher scores for sensorial attributes compared to the control. For the product to be accepted by consumers, color and appearance must be uniform. The incorporation of BCP powder imparted a reddish-purple color (Figure 4) to the yogurts which was liked very much by the panelists. Yoghurts with 2% BCP powder showed the highest liking scores for its appearance and color therefore, the addition of BCP powder had a significant effect on the acceptability and desirability of the color of the samples.

One of the key aspects of food products that influences consumer acceptance is texture [63]. BCP powder-containing yogurts were scored higher by sensory evaluators because they were more consistent and had a better texture than other samples. They also demonstrated a lower syneresis than plain yogurts in this regard.

The results of the taste sensory evaluation showed that the enriched yogurts had a significantly better taste than the plain yogurt sample. The addition of BCP at 2% did significantly change the taste of the yogurt (*p* < 0.05). In the same way, there was a discernible difference in the odor and flavor attribute scores across the yogurts. In general, the flavor and odor scores were still above the acceptable level, in which a score of >8 was obtained. The yogurts with added sea black carrot powder were evaluated as having a balanced taste, odor, and flavor.

Yogurt containing 1% and 2% BCP powder demonstrated higher scores for sensory parameters compared to the control. According to Figure 3, the yogurt sample with a 2% BCP content proved more satisfactory compared to the samples with 1% BCP. All sensory properties achieved the highest degree of acceptance at this percentage. All of the proposed samples were positively evaluated by the panelists, with no black carrot flavor being perceived. According to Elm Samh et al. [64], the probiotic yogurt sample that had 1.5%, 0.5%, and 1.5% black carrot jam added was the most well-liked. Additionally, Madora et al. [43] found that adding 1% and 2% carrot powder to yogurts is safe for the consumer’s health.

Overall, it was observed that all the samples of yogurt enriched with black carrot powder exhibited favorable sensory evaluations. The panelists found that adding 2% BCP to yogurt was the most pleasant addition and that it produced a better result than other yogurt products that had been treated.

## 4. Conclusions

The results highlighted that the BCP extract is an important source of bioactive compounds with high antioxidant activity. This investigation showed that adding BCP powder improves yogurt’s bioactive content and antioxidant capacity, which in turn improves yogurt’s nutritional profile. The sensorial analysis revealed that panelists appreciated the improved color of the yogurt samples. An overall appreciation of the value-added yogurt was noticed. After analysis, it was concluded that the yogurt sample (YBCP2) with 2% BCP powder had the best formulation. Compared to the yogurt without BCP powder, the samples supplemented with BCP powder had significantly increased amounts of minerals (Mg, P, K, Ca, and Na) because dried raw material BCP powder was added to the yogurt’s formulation.

The rich nutritional content of dried BCP powder (anthocyanin, polyphenols, and minerals), will be beneficial when added to yogurts, developing novel dairy products that are both highly appealing and naturally nutritious for consumer use. By utilizing by-products derived from the industrial processing of black carrots, it is feasible to explore their potential as a substitute for synthetic colors and a source of antioxidants. In addition to supporting the implementation of a circular economy model for environmental preservation, this can contribute to the reduction of waste in the food industry and serve multiple purposes.

## Figures and Tables

**Figure 1 foods-13-01130-f001:**
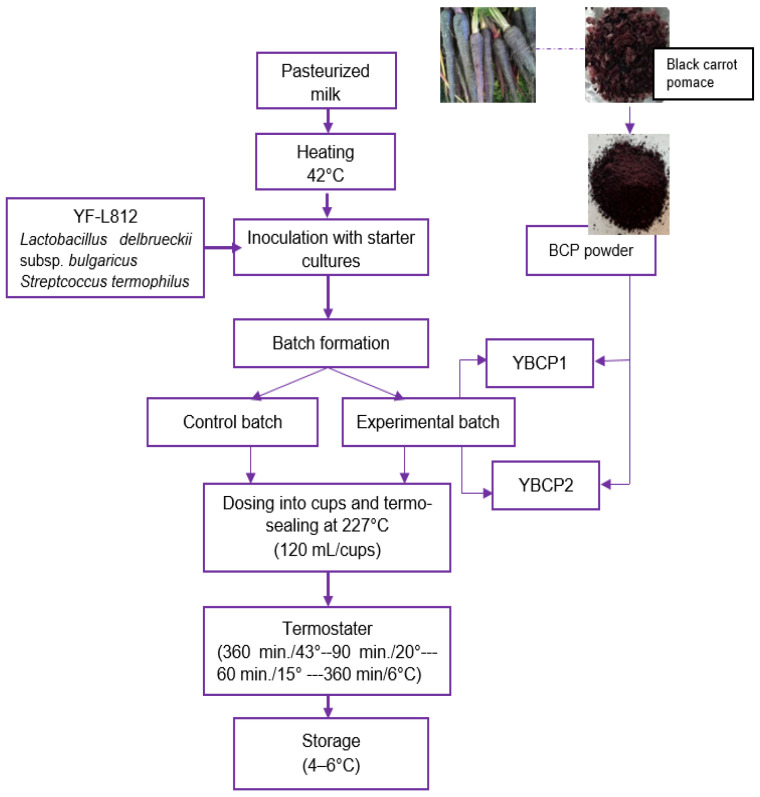
Flow chart of yogurt production.

**Figure 2 foods-13-01130-f002:**
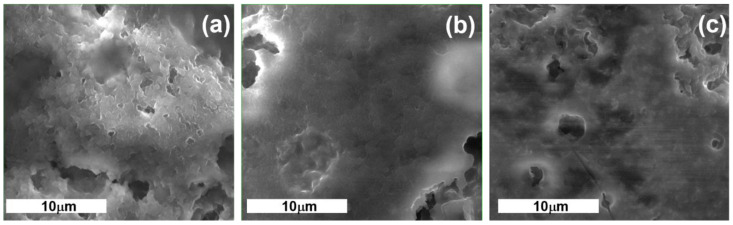
SEM micrographs of yogurt enriched with BCP powder: (**a**) YC yogurt without powder addition; (**b**) YBCP1 and (**c**) YBCP2 yogurt with 1 and 2% of BCP powder, respectively.

**Figure 3 foods-13-01130-f003:**
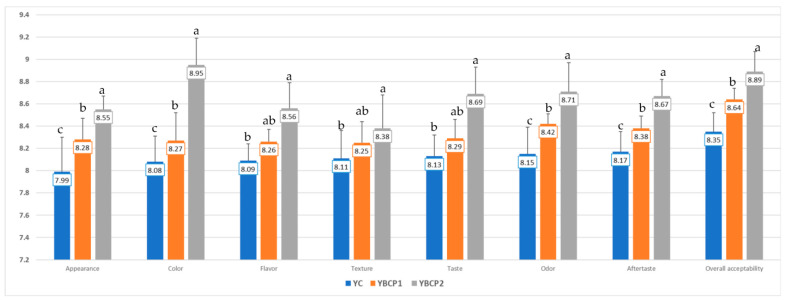
Comparative diagram of the sensory attributes specific to supplemented yogurts (averages with different letters “a”, “b”, and “c” in the columns signify statistically significant differences (*p* < 0.05)). YC—yogurt without powder addition; YBCP1 and YBCP2—yogurt with 1 and 2% powder of BCP.

**Figure 4 foods-13-01130-f004:**
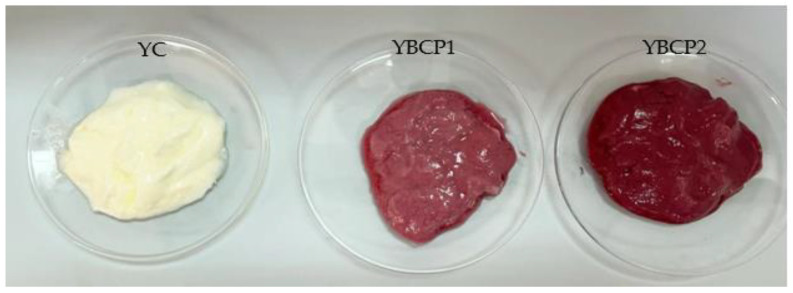
Yogurt samples with different percentages of BCP powder: YC (control), 1% (YBCP1), and 2% (YBCP2).

**Table 1 foods-13-01130-t001:** Phytochemical and color proprieties of BCP powder.

Parameter	BCP Powder
Total anthocyanins, mg C3G/g dw	3.46 ± 0.34
Total flavonoids, mg CE/g dw	18.49 ± 0.86
Total polyphenols, mg GAE/g dw	22.89 ± 0.87
Antioxidant activity,µmol TE/g dw	24.19 ± 0.66
Inhibition, %	81.85 ± 1.02
L*	21.29 ± 0.11
a*	8.17 ± 0.13
b*	−1.01 ± 0.03
Chroma	8.23 ± 0.13
Hue angle	359.88 ± 0.01
ΔE	22.82 ± 0.14
Calcium (Ca, mg/100 g)	176.75 ± 0.26
Phosphorus (P, mg/100 g)	58.81 ± 0.21
Potassium (K, mg/100 g)	131.05 ± 0.24
Magnesium (Mg, mg/100 g)	101.62 ± 0.25
Zinc (Zn, mg/100 g)	7.62 ± 0.22
Iron (Fe, mg/100 g)	9.67 ± 0.19
Sodium (Na, mg/100 g)	112.8 ± 0.31

**Table 2 foods-13-01130-t002:** Phytochemical characteristics and antioxidant activity of control and supplemented yoghurts and stability during 21 days of storage.

Phytochemical Characteristics	Storage Time, (Days)	YC	YBCP1 (1%)	YBCP2 (2%)
Total anthocyanins, mg C3G/100 g dw	0	-	9.15 ± 0.24 ^aA^	18.06 ± 0.42 ^aB^
	21	-	7.82 ± 0.21 ^bA^	15.61 ± 0.38 ^bB^
Total polyphenols mg GAE/g dw	0	2.26 ± 0.11 ^aA^	4.86 ± 0.05 ^aB^	6.97 ± 0.07 ^aC^
	21	1.21 ± 0.10 ^aA^	3.13± 0.06 ^aB^	5.22 ± 0.08 ^aC^
Total flavonoids mg CE/g dw	0	1.95 ± 0.07 ^aA^	3.53 ± 0.08 ^aB^	5.75 ± 0.09 ^aC^
	21	1.01 ± 0.06 ^aA^	2.05 ± 0.07 ^aB^	4.04 ± 0.07 ^aC^
Antioxidant activity, µmol TE/g dw	0	8.77 ± 0.38 ^aA^	17.15 ± 0.31 ^aB^	24.15 ± 0.26 ^aC^
	21	6.31 ± 0.25 ^bA^	15.09 ± 0.27 ^bB^	22.11 ± 0.21 ^BC^

Mean values followed by different superscript lowercase letters in the same column are significantly different (*p* < 0.05). Mean values followed by different superscript uppercase letters in the same raw are significantly different (*p* < 0.05).

**Table 3 foods-13-01130-t003:** Physico-chemical characteristics of control and supplemented yogurt during cold storage for 21 days.

Physical-Chemical Characteristics	Storage Time (Day)	YC	YBCP1 (1%)	YBCP2 (2%)
Total solids, %	0	14.93 ± 0.07 ^cA^	16.09 ± 0.02 ^bB^	17.04 ± 0.03 ^aB^
	21	14.43 ± 0.07 ^cB^	16.49 ± 0.01 ^bA^	17.26 ± 0.03 ^aA^
pH	0	4.59 ± 0.006 ^aA^	4.43 ± 0.01 ^bA^	4.39 ± 0.01 ^cA^
	21	4.38 ± 0.007 ^aB^	4.34 ± 0.01 ^bB^	4.31 ± 0.01 ^bB^
Fat, %	0	3.82 ± 0.03 ^cA^	3.95 ± 0.02 ^bA^	4.14 ± 0.02 ^aA^
	21	3.81 ± 0.03 ^bA^	3.65 ± 0.02 ^cB^	4.11 ± 0.02 ^aA^
Protein, %	0	4.30 ± 0.18 ^bA^	4.68 ± 0.04 ^bA^	5.11 ± 0.03 ^aA^
	21	4.29 ± 0.18 ^bA^	4.38 ± 0.04 ^bB^	4.82 ± 0.03 ^aB^
Carbohydrates, %	0	5.99 ± 0.22 ^aA^	4.71 ± 0.03 ^bB^	3.96 ± 0.03 ^cB^
	21	5.75 ± 0.21 ^aA^	5.74 ± 0.03 ^aA^	4.54 ± 0.06 ^bA^
Fiber, %	0	0.00 ± 0.00 ^cA^	1.69 ± 0.02 ^bA^	2.11 ± 0.01 ^aA^
	21	0.00 ± 0.00 ^cA^	1.67 ± 0.02 ^bA^	2.09 ± 0.01 ^aB^
Humidity, %	0	85.07 ± 0.07 ^aB^	83.91 ± 0.01 ^bA^	82.96 ± 0.03 ^cA^
	21	85.57 ± 0.07 ^aA^	83.51 ± 0.0 ^bB^	82.74 ± 0.03 ^cB^
Ash, %	0	0.82 ± 0.02 ^cA^	1.06 ± 0.01 ^bA^	1.71 ± 0.03 ^aA^
	21	0.57 ± 0.02 ^cB^	1.05 ± 0.01 ^bA^	1.69 ± 0.03 ^aA^
Energetic value, Kcal/100 g	0	75.09 ± 0.33 ^aA^	72.93 ± 0.13 ^cA^	73.53 ± 0.06 ^bB^
	21	74.09 ± 0.22 ^aB^	73.00 ± 0.13 ^bA^	74.27 ± 0.09 ^aA^
Syneresis, %	0	18.70 ± 0.66 ^aB^	13.40 ± 0.05 ^bB^	10.20 ± 0.06 ^cB^
	21	22.80 ± 0.22 ^aA^	20.40 ± 0.05 ^bA^	14.20 ± 0.06 ^cA^

For each physicochemical parameter and sample, values that do not share the same superscript lowercase letter are significantly different at *p* < 0.05. Samples that for each physicochemical parameter of each sample and storage time do not share the same superscript uppercase letter are significantly different at *p* < 0.05.

**Table 4 foods-13-01130-t004:** Mineral composition of different yoghurt samples.

Parameter	YC	YBCP1	YBCP2
Calcium (Ca, mg/100 g)	171.43 ± 0.39 ^c^	181.71 ± 0.44 ^b^	195.25 ± 0.34 ^a^
Phosphorus (P, mg/100 g)	54.08 ± 0.64 ^c^	63.22 ± 0.41 ^b^	71.56 ± 0.39 ^a^
Potassium (K, mg/100 g)	109.75 ± 0.86 ^c^	129.97 ± 0.86 ^b^	139.82 ± 0.99 ^a^
Magnesium (Mg, mg/100 g)	42.95 ± 0.57 ^c^	52.40 ± 0.65 ^b^	61.03 ± 0.55 ^a^
Zinc (Zn, mg/100 g)	2.71 ± 0.66 ^b^	3.41 ± 0.36 ^a^	3.49 ± 0.31 ^a^
Iron (Fe, mg/100 g)	0.21 ± 0.00 ^b^	4.79 ± 0.82 ^a^	4.85 ± 0.79 ^a^
Sodium (Na, mg/100 g)	82.33 ± 0.24 ^b^	106.12 ± 0.11 ^a^	106.23 ± 0.15 ^a^

Superscripts with different letters within a row are significantly (*p* < 0.05) different.

**Table 5 foods-13-01130-t005:** Colorimetric attributes of control yogurt and yogurts enriched with BCP powder during cold storage for 21 days.

Samples	Storage Time (Day)	L*	a*	b*	Chroma	Hue Angle	ΔE
YC	0	95.19 ± 0.26 ^aC^	−2.64 ± 0.03 ^aA^	18.54 ± 0.22 ^aB^	18.73 ± 0.22 ^aA^	178.57 ± 0.01 ^aB^	97.01 ± 0.28 ^aC^
	21	94.61 ± 0.32 ^bC^	−2.03 ± 0.04 ^bA^	19.08 ± 0.12 ^aB^	19.19 ± 0.11 ^bA^	178.53 ± 0.02 ^aB^	96.54 ± 0.24 ^aC^
YBCP1 (1%)	0	58.62 ± 0.13 ^aA^	27.60 ± 0.24 ^aC^	−2.43 ± 0.06 ^aA^	27.71 ± 0.24 ^aC^	359.91 ± 0.01 ^aA^	64.84 ± 0.14 ^aA^
	21	56.42 ± 0.14 ^bA^	29.51 ± 0.31 ^bC^	−2.82 ± 0.07 ^bA^	29.64 ± 0.22 ^bC^	359.90 ± 0.02 ^bA^	63.73 ± 0.12 ^bA^
YBCP2 (2%)	0	48.23 ± 0.93 ^aB^	32.52 ± 0.11 ^aB^	−2.01 ± 0.01 ^aB^	32.58 ± 0.11 ^aB^	359.95 ± 0.01 ^aB^	58.20 ± 0.52 ^aB^
	21	45.91 ± 0.81 ^bB^	35.21 ± 0.68 ^bB^	−1.59 ± 0.05 ^bC^	35.24 ± 0.13 ^bB^	359.94 ± 0.02 ^bB^	57.87 ± 0.59 ^aB^

Color parameter variation over time is highlighted by small letters. The color differences between the samples are highlighted by capital letters. Values that share a lower/uppercase letter are not significantly different (*p* > 0.05).

## Data Availability

The original contributions presented in the study are included in the article, further inquiries can be directed to the corresponding authors.

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
