# Peer review of "Application of Pomace Powder of Black Carrot as a Natural Food Ingredient in Yoghurt"

_foods, 2024, doi:10.3390/foods13071130_

Round 1

Reviewer 1 Report

Comments and Suggestions for Authors

The authors of the manuscript "Application of Pomace Powder of Black Carrot as a Natural Food Ingredient in Yoghurt" discussed the impact of adding freeze-dried black carrot pomace to yogurts in order to increase their health-promoting properties. The research concept is formulated and implemented correctly, although it is difficult to find any research novelties in it. In the discussion, the authors attached too much importance to discussing the differences between yogurts with two different doses of freeze-dried pomace (YBCP1 and YBCP2), where 1 and 2% additives were used, respectively, instead of paying more attention to the differences between natural YC yogurt and yogurts with additives. Comments on the text have been marked in the uploaded file. The article requires some corrections.

Author Response

Thank you for your observation. The manuscript has been revised.

Reviewer 2 Report

Comments and Suggestions for Authors

Dear authors. 

The manuscript showed original and relevant information. However, several comments need to be considered: 

Introduction

Information about black carrot byproducts must be included and developed, such as in the abstract. Mention the amount of byproducts generated and their traditional usage. 

Results and discussion

Line 286. Change the sentence powerful antioxidants to antioxidants.

Line 291-292. You mentioned that the evaluated treatment possesses high levels. Is this affirmation based on one classification of natural products? Please include this information. 

Line 332. Included a classification to establish that one treated possesses exceptional flavonoid content. 

Section 3.3. I believe that the discussion of each variable evaluated needs to be improved, as it was in the previous sections.

Conclusion. I suggest summarizing the main ideas of this section. 

Author Response

(The authors gave the same response as above.)
